# Applications of the CRISPR/Cas9 System for Rice Grain Quality Improvement: Perspectives and Opportunities

**DOI:** 10.3390/ijms20040888

**Published:** 2019-02-19

**Authors:** Sajid Fiaz, Shakeel Ahmad, Mehmood Ali Noor, Xiukang Wang, Afifa Younas, Aamir Riaz, Adeel Riaz, Fahad Ali

**Affiliations:** 1State Key Laboratory of Rice Biology, China National Rice Research Institute, Hangzhou 310006, China; shakeelpbg@gmail.com (S.A.); aamirriaz33@gmail.com (A.R.); fahadali.pbg@hotmail.com (F.A.); 2Institute of Crop Sciences, Chinese Academy of Agricultural Sciences, Key Laboratory of Crop Physiology and Ecology, Ministry of Agriculture, Beijing 100081, China; mehmood2017@gmail.com; 3College of Life Sciences, Yan’an University, Yan’an 716000, Shaanxi, China; 4Department of Botany, Lahore College for Women University, Lahore 54000, Pakistan; afifa_younas@hotmail.com; 5Biotechnology Research Institute, Chinese Academy of Agricultural Sciences, Beijing 100081, China; adeelpopy@yahoo.com

**Keywords:** functional genomics, *Oryza sativa* L. molecular markers, reverse genetics, CRISPR/Cas9

## Abstract

Grain quality improvement is a key target for rice breeders, along with yield. It is a multigenic trait that is simultaneously influenced by many factors. Over the past few decades, breeding for semi-dwarf cultivars and hybrids has significantly contributed to the attainment of high yield demands but reduced grain quality, which thus needs the attention of researchers. The availability of rice genome sequences has facilitated gene discovery, targeted mutagenesis, and revealed functional aspects of rice grain quality attributes. Some success has been achieved through the application of molecular markers to understand the genetic mechanisms for better rice grain quality; however, researchers have opted for novel strategies. Genomic alteration employing genome editing technologies (GETs) like clustered regularly interspaced short palindromic repeats (CRISPR)/CRISPR-associated protein 9 (Cas9) for reverse genetics has opened new avenues of research in the life sciences, including for rice grain quality improvement. Currently, CRISPR/Cas9 technology is widely used by researchers for genome editing to achieve the desired biological objectives, because of its simple targeting. Over the past few years many genes that are related to various aspects of rice grain quality have been successfully edited via CRISPR/Cas9 technology. Interestingly, studies on functional genomics at larger scales have become possible because of the availability of GETs. In this review, we discuss the progress made in rice by employing the CRISPR/Cas9 editing system and its eminent applications. We also elaborate possible future avenues of research with this system, and our understanding regarding the biological mechanism of rice grain quality improvement.

## 1. Introduction

Rice (*Oryza sativa* L.) feeds more than 3.5 billion people worldwide [1]. Rice grain quality preferences differ between geographical regions and/or ethnic groups [2]. The prime characteristics affecting quality are cooking and eating parameters, phytochemicals, and micronutrients [3]. The assessment of grain quality is a laborious and time-consuming task that not only requires large amounts of samples at early stages of breeding, but also adherence to standard protocols [4]. Understanding the molecular basis of grain quality has been a prime objective of past quality improvement research. The fine-mapping and cloning of quantitative trait loci (QTLs) for grain quality improvement have received more attention recently, because of its large economic value and consumer preference [5]. Several regulatory and structural genes, chemical pathways, and regulatory networks involved in grain quality, have been identified using various approaches. In the current scenario, conventional mutational breeding techniques, i.e., ethyl methanesulfonate and X-rays, have multiple limitations, and new techniques, i.e., CRISPR/Cas9 are highly desirable for achieving the goal of rice grain quality improvement with more precision and higher efficiency [6]. 

The functions of various components of the rice genome ultimately enable the production of higher yielding varieties with better quality rice grains [7,8]. Currently, CRISPR/Cas9 is a widely adopted genome editing technology (GET) because of its simplicity, efficiency, and versatility [9]. The target specificity in CRISPR-Cas9 system is the most reliable, as target sites are recognized by Watson and Crick model, and off-target sites are identified through sequence analysis [10]. CRISPR/Cas9 cleaves foreign DNA via two components, Cas9 and single guide RNA (sgRNA). Cas9 is a DNA endonuclease that can be derived from different bacteria, such as *Brevibacillus laterosporus* [11], *Staphylococcus aureus* [12], *Streptococcus pyogenes* [13], *Streptococcus thermophilus* [14], and *Streptococcus pyogenes* is the most widely used for Cas9 isolation. Cas9 contains two domains, i.e., the HNH domain and the RucV-like domain. The HNH domain cuts the complementary strand of CRISPR RNA (crRNA), while the RucV-like domain cleaves the opposite strand of the double-stranded DNA. The sgRNA is a synthetic RNA with a length of about 100 nt. Its 5’-end has a 20-nt sequence that acts as a guide sequence to identify the target sequence accompanied by a protospacer adjacent motif (PAM) sequence, which is often the consensus NGG (N, any nucleotide; G, guanine). The loop structure at the 3’-end of the sgRNA can anchor the target sequence by the guide sequence and form a complex with Cas9, which cleaves the double stranded DNA and forms a double-strand break (DSB) at this site. Once a DSB is generated, nonhomologous end-joining (NHEJ) or homology-directed repair (HDR) DNA repair mechanisms are initiated. A DSB is usually repaired by NHEJ in most situations, and it is a simple way to create mismatches and gene insertion/deletions (indel), leading to gene knockout. When an oligo-template is present, HDR induces specific gene replacement or foreign DNA knock-ins [15,16]. These processes are all ways in which CRISPR/Cas9 can efficiently edit the genome of diverse organisms, including humans, animals and plants (Figure 1). 

Recent reviews have described the genetics and biotechnologies [17], integration of knowledge from omics-based studies [18], and methods for utilizing genome editing, particularly CRISPR/Cas9 for improving rice grain quality [19,20]. Scientists equipped with CRISPR expertise have contributed by disseminating relevant information to CRISPR newcomers, in contrast to the proprietary nature of zinc-finger nucleases. In addition, several online platforms are now available to assist researchers with all concerns relating to CRISPR [21,22,23,24,25,26]. Based on these developments, we provide a non-comprehensive review with special emphasis on the applications of the CRISPR/Cas9 system for the development of rice varieties with better grain quality.

## 2. Genetics and Genomics of Rice Grain Quality

Milling quality is based on the recovery of brown, milled, and head rice. It is a complex grain characteristic whose genetics are not fully understood. Over recent years, high throughput mapping technologies have enabled the identification of several major QTLs associated with rice quality. Two major QTLs (*qBRR3*, *qBRR5*), were mapped to chromosomes five and three, and these influence brown rice recovery [27,28]. The QTLs *qBRR3* and *qBRR5* were also found to regulate grain width and length [28]. Another study mapped the major QTL *qHRR3*, located on chromosome three, and it is associated with head rice recovery and grain length [27,29]. Based on these studies, there is a strong link between grain size or shape and the percentage of head rice recovery. Consistently, four QTLs related to grain size, including *GW2* [30], *GW5* [31], *GW8* [32], and *GS5* [31], have also been cloned and functionally characterized. Concomitantly, their impact on cooking and eating qualities were also established. The *Waxy* (*Wx*) gene, on chromosome six, is an important gene which controls amylose content (AC), gel consistency (GC), and rapid visco analyzer pasting viscosity [33]. Additionally, many Wx alleles are present in different AC classes. Five prevalent classes of AC, glutinous, low, intermediate, high I, and high II contain five common alleles: *Wx*, *Wx^t^*, *Wxg^1^*, *Wxg^2^*, and *Wxg^3^*, respectively [34]. Additionally, chalkiness is an integral component that determines the quality and ultimately the economic value of rice grain. Chalkiness negatively impacts appearance, milling, cooking, and nutritional qualities, as well as head rice recovery [35]. Chalkiness is largely determined by external and internal cues. 

For example, rice cultivated at high temperatures has higher chalkiness, and genes involved in starch biosynthesis, grain filling, and starch granule structure all hamper chalkiness [36,37]. Based on these observations, genes controlling grain chalkiness, including *qPGWC-8* [38] and *qPGWC-7* [39], have been fine-mapped. However, the mechanism underlying the formation of grain chalkiness remains elusive.

Gelatinization temperature (GT) and amylopectin structure are controlled by a major QTL, SSIIa, which is located on chromosome six [33,40,41]. However, starch biosynthesis-related genes, such as *SBE1*, *BE3*, *AGPlar*, *PUL*, *ISA*, and starch synthesis genes, i.e., *SSI*, *SSIIa*, *SSIII-2*, and *SSIV-2* all have effects on cooking and eating quality [42,43]. Starch biosynthesis pathways and genes/enzymes in rice endosperm are well characterized (Figure 2). A deletion within exon seven [44] or exon two [45] of the *BADH2* gene, located on chromosome eight, increases the level of 2-acetyl-1-pyrroline (2AP) in fragrant rice. These mutations render *BADH2* non-functional, resulting in increased 2AP [44]. However, the genetics and biochemical pathways of fragrance need further investigation [3]. In addition, a major QTL for protein content in rice, *qPC1*, has been cloned and functionally characterized [46]. A major QTL responsible for crude fat content (FC) in brown rice, *qCFC5*, is located on chromosome five [47]. QTLs controlling FC are governed by time-dependent gene expression; Wang et al. [48] revealed different expression patterns of FC-related QTLs at the grain filling stage. The possible mechanisms and QTLs responsible for amino acid content have been characterized by many researchers [49,50]. In addition, several QTLs responsible for mineral accumulation in rice have been characterized [51]. Based on these studies, QTLs controlling mineral accumulation in rice grain are largely environmentally regulated [52].

Traditionally studies on improving rice grain quality through genetic control have been conducted using biparental mapping populations, whilst the latest techniques of genome-wide association studies (GWASs) have allowed the understanding of the genetic basis of complex traits, i.e., grain quality. Although a sufficient number of studies [53,54,55,56,57] have been carried out using GWAS, and various genes/QTLs has been identified as well, which are associated with important grain quality parameters, further characterization of the identified candidates is needed. With the onset of next-generation sequencing, the construction of a high-resolution genetic map has become handy for to analyzing population genetics and for expression analysis. By employing this sequencing approach, Chen at al. [58] had constructed a high-density genetic map for a RIL population having 2711 recombination bin markers. They detected 12 QTL clusters, four of which matched the genomic regions of cloned genes or fine-mapped QTLs, i.e., *GL7* [59], *GS3* [60], *gw5/qSW5* [61,62], and *qPGWC-7* [63]. Besides that, eight other novel QTL clusters for grain shape and chalkiness were obtained [58]. The integration of the various -omics approaches, including genomics, transcriptomics, proteomics, metabolomics, etc., which can be termed as “multi-omics”, may exploit underlying mechanisms to improve rice grain quality traits by understanding pathways for seed development and grain quality attributes [18], which definitely require extra capital and resources from other breeding platforms, i.e. bioinformatics. Based on the presented facts and the well-documented functional genomics of rice grain, along with the availability of genetic resources and the high transformation efficiency, the employment of the CRISPR/Cas9 system is a better choice for rice grain quality improvement.

## 3. Applications of CRISPR/Cas9 for Rice Grain Quality Improvement

The improvement of rice grain quality attributes via targeted genome editing is a fast, sustainable, and cost-effective approach. The application of CRISPR/Cas9 requires multiple processes. The initial step is to discover genes of significant importance. The genes that negatively regulate the grain quality, can be referred to as Q-genes (any plant gene that facilitates the degradation process of rice grain quality when expressed). Both forward and reverse genetics approaches can identify the genes that are responsible for phenotypic variation [3]. Conventional plant breeding tools mainly depend on naturally existing germplasm variations. The introgression of desirable traits into the selected germplasm requires successive backcrossing, followed by the screening of large populations, which requires much time and energy. However, reverse genetic approaches enhance the speed of plant breeding through targeted genome modification (Figure 3). The available literature on CRISPR/Cas9 has promoted its application towards the genetic improvement of *O. sativa* L. (Table 1, Figure 4).

### 3.1. Improving Rice Grain Appearance and Milling Quality

Many genes responsible for rice grain appearance quality have been identified, and have the potential to be tapped with CRISPR/Cas9 technology. Grain appearance is the primary quality attribute that influences the market acceptability of rice [17]. However, another important quality trait is the grain chalkiness, which is an undesirable quality attribute, which results in low market acceptability [78]. Grain shape is regarded as a yield component, and it plays a key role in determining the quality of rice grains. Recently, *GS3*, responsible for grain length, and *Gn1a*, controlling the number of grains, have been successfully edited in four rice varieties [79]. The transgene-free T_1_ plants showed longer grain lengths and increased thousand grain weights in comparison to the wild type. Similarly, three other important genes *GW2*, *GW5*, and *TGW6*, negative regulators of grain weight, were targeted through CRISPR/Cas9-mediated multiplex genome editing. The obtained results indicated that the genome editing of these genes significantly increased grain size and thousand grain weight [72]. There are several identified genes for these traits, and their interactions and functions have been fully characterized. The integration of functionally characterized genes/QTLs, i.e., *OsSPL13*, *OsSPL16/GW8*, *GW7*, and *Chalk5*, through knock-in/out using CRISPR/Cas9, and the assessment of their interactions among genes, can greatly improve our understanding of rice grain appearance and milling quality.

### 3.2. Improving Rice Grain Cooking and Eating Quality

Generally, three important physicochemical indices comprising amylose content (AC), gel consistency, and the gelatinization temperature (GT), altogether determine the cooking and eating quality of rice grain. All three indices define (~90%) the starch properties of hulled rice. Rice with good cooking and eating quality determines the cooking ease, along with the firmness and the stickiness features [80]. AC is regulated by the *Wx* gene in the endosperm [81], and *ALK/SSIIa* and *RSR1* control GT [82]. Several studies have been conducted to unearth the roles of different genes and/or enzymes involved in the regulation and expression of the *Wx* gene. Wu et al. [83] reported that dozens of dull genes affect the splicing efficiency of *Wx*, and they characterized a tetratricopeptide domain-containing flo2 protein regulating the expression of *Wx*. Similarly, transcription factors like *rsr1*, *OsBP-5*, *OsEBP-89*, *OsbZIP58*, and *OsMADS7* modified *Wx* expression [84,85]. The indica hybrids, especially in China, are high in AC, and become hard and dry during cooking. Ma et al. [86] successfully edited the *Wx* gene in the japonica background, leading to reduced AC. Moreover, transgenic Taichung 65 rice lines containing a *Wx* antisense construct had lower AC, and hybrids obtained from these transgenic lines also showed reduced AC [87]. In comparison, [64,84] introduced a loss-of-function mutation via CRISPR/Cas9 to the *Wx* gene in two widely grown japonica cultivars, “Xiushui134” and “Wuyunjing 7”. The *Wx* gene mutation led to reduced AC, and offered an effective strategy to improve elite cultivars without any penalty in other desirable agronomic traits. To understand the fine structure and physical properties of starch *SBEI* and *SBEII*, they were mutated via targeted mutagenesis. The results demonstrated that *SBEIIb* plays an important role in creating high-amylose rice. The fragrant gene *Badh2* in Zhonghua 11 was edited by the CRISPR/Cas9 mutagenesis system. The mutated line contained an additional T base in the first exon of *Badh2*, and resulted in an increased amount of 2AP, and enhanced fragrance in rice [88]. The successful editing of these genes has proven that this system could be the best method for understanding the functional aspects of genes and transcriptional factors influencing cooking and eating quality.

### 3.3. Improving Rice Grain of Nutritional Quality

Food nutritional quality improvement has great importance worldwide, especially in developing countries where many people rely on rice as their staple food. Additionally, approximately 24,000 people die daily globally, owing to malnutrition [89,90]. People are eating foods that are deficient in protein, energy, iron, zinc, vitamin A, and iodine [91]. However, improving rice grain nutritional quality using CRISPR/Cas9 system can overcome this issue. The content of seed storage proteins (SSPs), fats, amino acids, vitamins, and other micronutrients determines the nutritional quality of rice grain [46], but rice has the lowest protein content of the cereal grains [92]. However, the protein is the second most abundant ingredient of hulled rice after starch contents, in that the lysine is amongst the limiting essential amino acids, as per human nutrition standards [17]. Certain amino acids such as lysine (Lys) and tryptophan (Trp) are missing from the SSPs of rice grain [93,94]. Therefore, improving nutrition in humans is intriguingly associated with improving nutrition in SSPs, especially for people in regions where rice is a staple food. Rice contains six 5-methylcytosine (5mC) DNA methylase genes, including *OsROS1*, *OsROS1b*, *OsROS1c*, *OsROS1d*, *OsDML3a*, and *OsDML3b*, which can play an important role in enhancing nutritional grain quality. The number of aleurone cell layers was increased by *ta2-1*, which is a weak mutant allele of *OsROS1* [95]. Approaches to improve protein and essential amino acids in rice seeds by transgenic engineering have been attempted by many research groups, e.g., the expression of AmA1 seed albumin [96], the overexpression of aspartate aminotransferase genes [39], the transfer of two artificially synthesized genes [97], the production of engineered rice [68], the transfer of wheat glutelin gene *Glu-1D × 5* [98], and the expression of a gene encoding a precursor polypeptide of sesame 2S albumin [99].

The SSPs include four different categories in rice, albumin, globulin, prolamin, and glutelin, separated by their solubility [100]. In rice, SSP genes have been cloned and characterized mostly by mutant screening [101]. Movement of the ferritin gene from common bean into rice has been made possible by transgenic approaches [102]. The concentration of Fe in ferritin-containing rice lines was double the concentration in controls. In addition, Vasconcelos et al. [103] transferred the *ferritin* gene from soybean into rice, and recorded Fe concentrations. Interestingly, it was increased by three- and two-fold in milled and rough rice, respectively. Similarly, Khalekuzzaman et al. [104] introduced the *ferritin* gene driven by an endosperm-specific glutelin promoter, and found increased Fe concentration in brown and polished seeds of T_1_ and T_2_ populations of the cultivar, BRRl Dhan 29 (BR29), respectively, in comparison with controls. Thus, the Fe content was increased by over two-fold in transgenic lines. Subsequently, many researchers have attempted to increase Fe content in rice endosperm using different methods such as: (1) overexpressing the genes involved in Fe uptake from the soil; (2) moving Fe into the grain from the root, shoot and flag leaf; and (3) increasing the efficiency of Fe storage proteins [105,106,107]. The *nicotianamine synthase* (*NAS*) gene from *Hordeum vulgare* L. has been successfully transferred into rice, which significantly improved the contents of Fe and Zn by two- or three-fold in polished rice grain. In another study, Zheng et al. [108] observed that the overexpression of endosperm-specific endogenous *NAS* genes (*OsNAS1*, *OsNAS2*, and *OsNAS3*) increased Fe accumulation by five-fold in polished rice grain. In addition, Johnson et al. [109] recorded a two-fold increase of Fe and Zn concentration in polished rice overexpressing single rice *OsNAS* genes. β-carotene, which is the precursor of vitamin A, cannot be produced by rice. Therefore, researchers developed golden rice, which rich in β-carotene, by the introgression of two genes, including *phytoene synthase* and *phytoene desaturase*, to overcome night blindness caused by vitamin A deficiency in many developing countries [110]. Briefly, all of these genes have been manipulated/introgressed from different biological backgrounds approaches.

However, these approaches have some limitations, such as being time consuming, involving the introduction of foreign DNA, off-target genome modifications, the association of undesirable traits with target attributes, and the lower efficiency makes them a hard choice for researchers. However, improving rice grain nutritional quality using the CRISPR/Cas9 system can overcome these issues. The CRISPR/Cas9 system was used to knock out five rice carotenoid catabolic genes (*OsCYP97A4*, *OsDSM2*, *OsCCD4a*, *OsCCD4b*, and *OsCCD7*), and to increase β-carotene accumulation in rice endosperm [111]. However, it was found that the targeted mutations in five carotenoid catabolism genes failed to boost carotenoid accumulation in rice seeds, which needs further investigations to make the following approach reliable. Multiplex editing is an easy and well-understood system, especially for comparing and dissecting the functions and relationships of major genes/QTLs [72]. In the rice genome, up to 46 target sites were edited, with an average of 85.4% mutation frequency [86]. The study also confirmed the immediate editing of three sites within the gene *OsWaxy*, which caused an amylose content reduction (up to 14%). Multiplex genome editing was also testified with the help of endogenous transfer RNA (tRNA) processing system in rice, wherever each sgRNA was flanked by tRNA and processed into single sgRNAs, which caused large deletions in the genomic sequences of the T_0_ generation [112]. Likewise, it a new strategy was reported in rice for the CRISPR/Cas9-sgRNA multiplex editing system, where 21 sgRNAs were designed, and the equivalent Cas9/sgRNAs expression vectors were created [113]. Transformed rice plants were successfully and significantly edited, and up to 82% of the desired target sites represented deletion, insertion, substitution, and inversion events, thus exhibiting high editing efficiency. All of these reports clearly show that CRISPR/Cas9 is the best method for rapidly validating the function of genes, and thereby testing many genes simultaneously. It can be used to establish proofs of concept before targeting the genes that are directly involved in the quality of the rice grain. 

## 4. Beyond Rice Grain Quality Improvement

Over the last few years, GETs have revolutionized crop improvement programs. Newly developed crop varieties have improved traits including high yield, resistance against different diseases, and biotic and abiotic stresses. Firstly, the knockout of genes that have a great influence on grain yield, such as *GS3*, *DEP1*, *GS5*, *GW2*, *Gnla*, and *TGW6*, is a simple and direct method for improving average rice yields. The mutants of these genes give the desired, impressive phenotypes [70,114]. The development of a rice triple mutant by simultaneously knocking out *GW2*, *GW5*, and *TGW6* increased the thousand grain weight significantly [72]. Moreover, hybrid rice production can also be increased by using *TMS5* mutant lines [115,116]. Multiple disease resistance lines have also been obtained via CRISPR/Cas9 technology. The knockout of the rice blast resistance gene *OsERF922* showed significant reduction in blast lesion formation under pathogen infection [74]. A knockout of the blast resistance gene *Bsrk-1* enhanced the resistance of rice without compromising yield [75]. Moreover, herbicide resistance research was initiated to ensure public and environmental health, as both are influenced by agrochemical use [117]. *ALS1* is one of the main enzymes responsible for the herbicide resistance of rice. Sun, Zhang, Wu, He, Ma, Hou, Guo, Du, Zhao and Xia [76] carried out mutations at multiple discrete points in the rice *ALS* gene using CRISPR/Cas9. The results showed that CRISPR/Cas9-mediated homology-directed repair was successful. Xu et al. [118] targeted the second exon of *BEL* in the Nipponbare rice cultivar, related to bentazon and sulfonylurea herbicide resistance, through CRISPR/Cas9. The phenotypic screening matched the results of the genetic mutant screening. Additionally, the seedling stage of rice is more vulnerable to low temperatures. *TIFY1b*, a transcription factor, and the *OsAnn3* gene in rice were edited through CRISPR/Cas9, which enhanced cold tolerance significantly. In addition, knockout of the *OsNramp5* transporter gene for cadmium (Cd) led to the development of rice hybrid lines with low Cd accumulation. The mutant osnramp5 showed less accumulation of Cd in roots, shoots, and seeds [69,119]. Hence, genome editing using the CRISPR/Cas9 system has contributed a lot to the manipulation of plant genomes, but there is still great potential to be tapped.

## 5. Conclusions and Future Perspectives

Premium quality rice grain is the demand of a growing population with better living standards. Presently, the CRISPR/Cas9 system has all genome editing capabilities, e.g., knock-in, knockout, knockdown, and expression activation. This system has tremendous untapped potential, has formed an ever-expanding genetic toolbox for plant biologists to investigate functional genomics, and is a helping hand for breeders to integrate important genes into the genomes of important crops. The successful application of CRISPR/Cas9 for tissue engineering and human stem cell modification has led to further developments in the field of precise genome editing. The ability to target multiple genes via multiplexed genome editing strategies can facilitate pathway-level research to engineer complex multigenic rice grain quality attributes. Previously, few studies have been conducted that are related to targeted mutagenesis for rice grain quality improvement. The pathways of rice grain quality are not well understood, and they can be investigated for the genetic mechanisms controlling quality attributes. The development of novel regulatory components from naturally existing peripherals (genes, promoters, *cis*-regulatory elements, small RNAs, and epigenetic modifications) can facilitate the engineering of regulatory pathways for different elements of rice grain quality. The rapid shift of research toward the utilization of CRISPR/Cas9 systems for targeted mutagenesis could be a promising approach for overcome barriers to breeding improved quality rice.

## Figures and Tables

**Figure 1 ijms-20-00888-f001:**
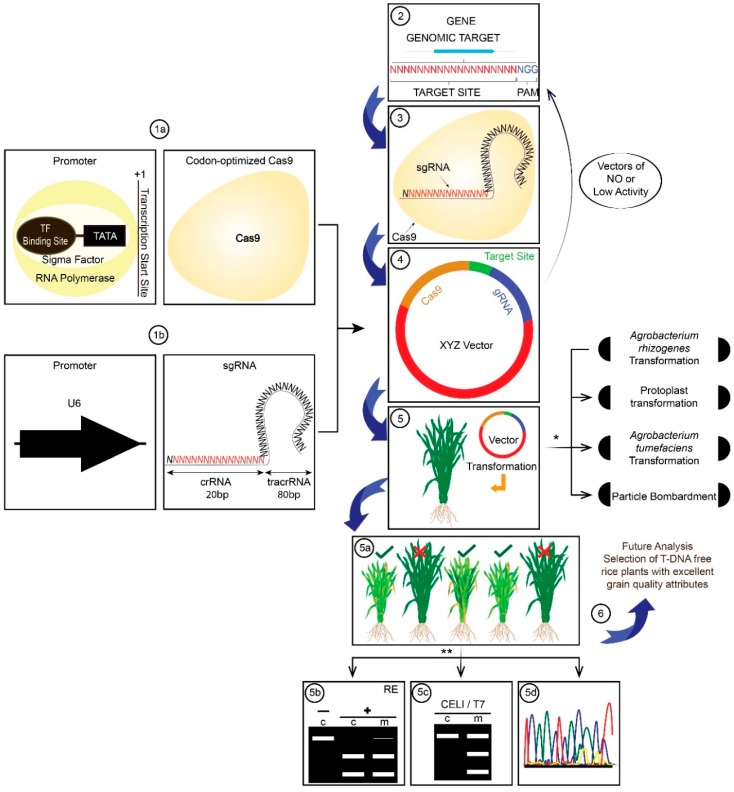
Basic flow chart of the CRISPR/Cas9 genome editing system. The engineered CRISPR/Cas9 system consist of two components; (**1a**) the Cas9 endonuclease and, (**1b**) a single-guide RNA (sgRNA). “The sgRNA contains a spacer sequence followed by 79 nt of an artificially fused tracrRNA and crRNA sequence”, (**2**) The spacer sequence is typically 20 nt in length, and specifically binds to the target DNA sequence containing a 5’-NGG-3’ PAM motif at the 3’ end, which is highly specific for the gene of interest, (**3**) The fused trans-activating crRNA (tracrRNA) and crRNA sequence forms a stem-loop RNA structure that binds to the Cas9 enzyme; tracrRNA hybridizes and joins Cas9. (**4**) Assembly of sgRNA, attached with the target sequence and the Cas9 vector construct. (**5**) Transformation of the vector construct into rice via different transformation techniques. (**5a**) Screening and selection of rice mutant plants based on phenotypic changes. (**5b**) Restriction enzyme site loss generating a CRISPR/Cas9 mutagenized plant line. (c, control; m, mutagenized; RE, restrictions enzyme). (**5c**) Surveyor Assay (CEL1 and T7 are DNA endonucleases utilized in surveyor assay). (**5d**) Next-generation sequencing. (**6**) Future analysis to obtain T-DNA-free plants, and further experiments to prove phenotypic changes cast by the knockout of the gene under investigation. * Different techniques for the vector construct transformation. ** Regeneration and screening of transgenic plants for gene editing events.

**Figure 2 ijms-20-00888-f002:**
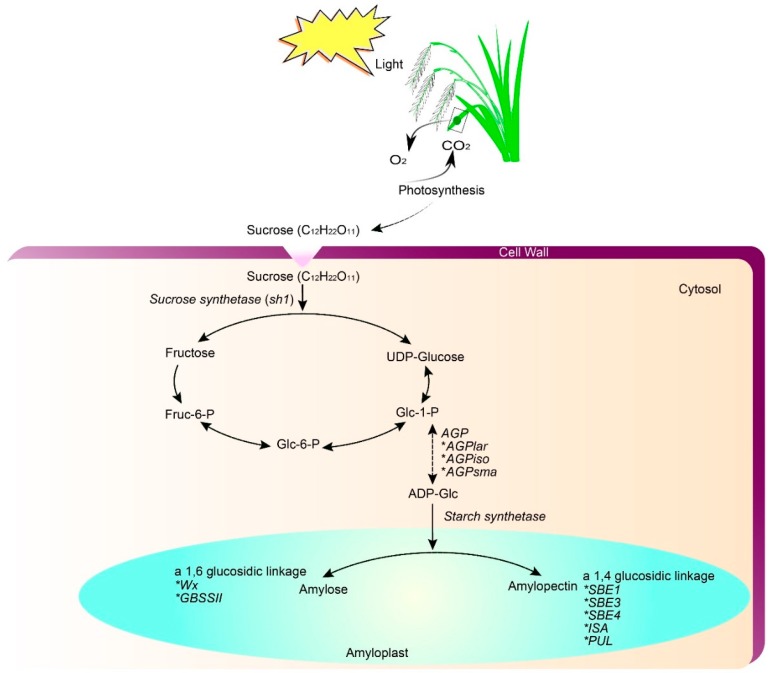
Starch biosynthesis pathway in cereals endosperm (modified from [18]). Eighteen genes play integral roles in different steps of starch synthesis. AGP, ADP-glucose pyrophosphorylase, AGPlar, AGP large subunit; AGPiso, AGP large subunit isoform; AGPsma, AGP small subunit; GBSS, granule bound starch synthase; SS, soluble starch synthase; SBE, starch branching enzyme; ISA, isoamylase; PUL, pullulanase; ISA and PUL belong to the starch debranching enzyme (DBE).

**Figure 3 ijms-20-00888-f003:**
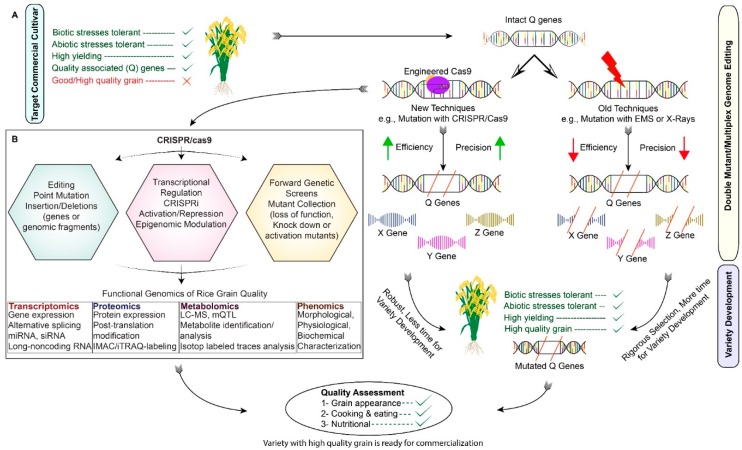
An illustration of rice grain quality improvement through the CRISPR/Cas9 system. (**A**) Advantages of CRISPR-mediated gene editing over conventional breeding techniques to develop rice varieties with high grain quality. (**B**) An overview of the applications of CRISPR-Cas9 system in the functional genomics of rice grain quality improvement. The CRISPR-Cas9 system can be used for genome editing (via the introduction of point mutations, insertions or deletions), transcriptional regulation (via CRISPRi (CRISPR interference), activation, repression, or epigenetic modulation) or forward genetics screens (via the generation of loss-of-function, knock-down, or activation mutants using sgRNA libraries) for understanding the molecular basis of rice grain quality, which can lead to the generation of crop plants with excellent quality grain.

**Figure 4 ijms-20-00888-f004:**
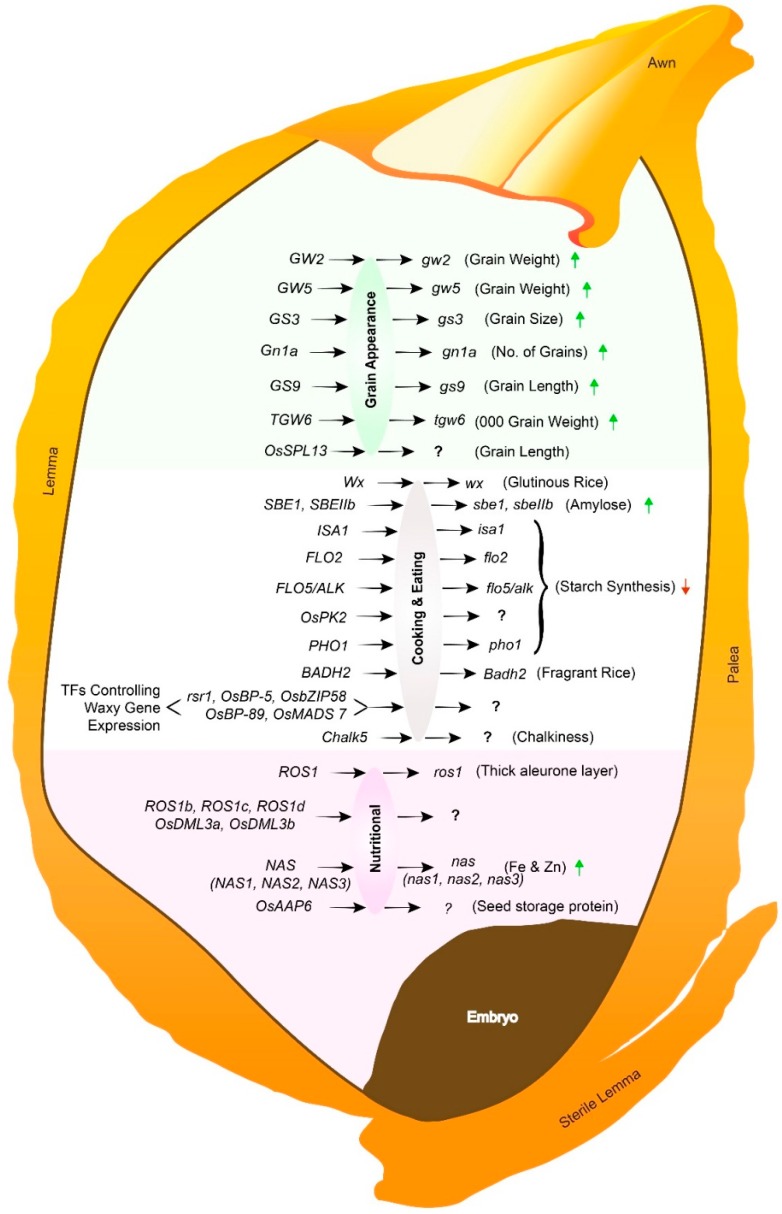
Genes responsible for rice grain quality, parallel to their mutated function. ? = potential genes for editing via CRISPR/Cas9 to improve the grain quality of rice varieties; red downward arrows (↓) represent a decrease in traits, whereas green upward arrows (↑) represent an increase/improvement in traits when their respective genes are mutated.

**Table 1 ijms-20-00888-t001:** CRISPR-Cas9-edited genes in *O. sativa* L.

Application Prospective	Target Gene	Cas9 Version	Cas9 Promoter	sgRNA Promoter	Transformation Method	Reference
Quality Improvement	*Waxy*	N/A	35S	OsU6	*Agrobacterium*-mediated transformation	[64]
	*SBE1, SBEIIb*	Codon-optimized Cas9	ZmUbi	OsU3		[65]
	*ISA1*	Rice codon-optimized	35S	OsU6		[66]
	*OsPDS, OsBADH2, Oso2g23823, OsMPK2*	Rice codon-optimized	2 × 35S	OsU3		[67]
	*OsCYP97A4, OsDSM2, OsCCD4a, OsCCD4b, and OsCCD7*	Rice codon-optimized	35S	OsU3		[68]
	*OsNramp5*	Rice codon-optimized	CaMV35S	OsU6a		[69]
Yield Improvement	*Gn1a, DEP1, GS3, IPA1*	Codon-optimized Cas9	OsUbi	OsU6a	*Agrobacterium*-mediated transformation	[70]
	*GLW2*	Plant codon-optimized	2 × 35S	OsU6		[70]
	*GS9*	Rice codon-optimized	CaMV 35S	OsU3		[71]
	*GW2, GW5 and TGW6*	Codon-optimized Cas9	OsUbi	OsU3, OsU6 and TaU3		[72]
	*TMS5*	Codon-optimized Cas9	OsU3/U6	OsU3/U6		[73]
Disease resistance	*OsERF922*	Codon-optimized Cas9	CaMV 35S	OsU6	*Agrobacterium*-mediated transformation	[74]
	*Bsrk1*	Rice codon-optimized	35S	OsU6		[75]
	*ALS*	Rice codon-optimized	2 × P35S	OsU6		[76]
	*OsAnn3*	Codon-optimized Cas9	CaMV35S	OsU6		[77]

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
