# Peer review of "Applications of the CRISPR/Cas9 System for Rice Grain Quality Improvement: Perspectives and Opportunities"

_ijms, 2019, doi:10.3390/ijms20040888_

Round 1
Reviewer 1 Report
In this review, Fiaz et al. list various approaches made to improve the nutritional and agronomical value of rice through classical transformation and through novel gene editing approaches. The subjects range from yield increase, increase of endosperm protein content and alteration of amino acid compositions, taste and increased micronutrient availability in endosperm (provitamin A, Fe). I have the following suggestions to improve the review:
1) Images
The principle of the CRISPR/Cas9 genome editing are touched only briefly which is reasonable in this review which focused more on its application. However, in order to understand the images, e.g. figure 1, the legend needs more explanation in such a way that the content of the image can be understood by reading the legend. The same applies for the other images and their legends as well. Moreover, images need to be better linked with the content of the chapters, e.g. most of the content in figure 1 does not appear in the text. Abbreviations need to be explained in the legend, e.g. what is CELI, what is c/c/m/-/+. Same for Figure 3: please include a legend with whole and descriptive sentences and link figure content with text.
2) Information
Information is somewhat spread over the manuscript in a repetitive way: e.g. the information on the BADH2 gene appears first in line 86 (page 2), then in line 44 (page 7), same applied for other genes. It is recommended to merge these information.
3) Golden Rice
Line 86: Golden rice was generated through introgression of only two genes: a phytoene synthase and a (bacterial) phytoene desaturase. Introgression of a lycopene cyclase is not required for b-carotene biosynthesis and a b-carotene desaturase does not exist. Paine et al. 2005 (Nat. Biotechn.) as an improvement should be cited. Please correct.
4) Title
Most of the genetic engineering approaches are from classical transformation approaches, not from gene editing. Often it is just mentioned that similar results were obtained. So, why CRISPR/Cas9 is the better choice – some more arguments and pointing towards convincing comparisons would be good!
Minor points
- Line 99: avoid subjective expressions (breath-taking)
Author Response
Dear editor,
Thank you very much for giving us time and comments regarding our manuscript. Our manuscript “CRISPR/Cas9 is a better choice for rice grain quality improvement: A review” has been revised carefully and here we have given the list of response to the reviewer’s comments. We have improved the manuscript according to the reviewer’s comments. All the revisions have been highlighted with green background color in the revised manuscript.
We are grateful to you and your team for your co-operation in advance.
With best regards,
Sajid Fiaz,
*********************************************************************
Response to reviewers:
Reviewer #1:
1. The principle of the CRISPR/Cas9 genome editing are touched only briefly which is reasonable in this review which focused more on its application. However, in order to understand the images, e.g. figure 1, the legend needs more explanation in such a way that the content of the image can be understood by reading the legend. The same applies for the other images and their legends as well. Moreover, images need to be better linked with the content of the chapters, e.g. most of the content in figure 1 does not appear in the text. Abbreviations need to be explained in the legend, e.g. what is CELI, what is c/c/m/-/+. Same for Figure 3: please include a legend with whole and descriptive sentences and link figure content with text.
Response: Thank you for your good suggestions. The all figures of MS have been improved as per your suggestion and detailed legends are given. Typography have been improved to make it readable. Moreover, the full names of abbreviations are given and figures are linked with the text. All of these changes are highlighted in red colour in text.
2. Information is somewhat spread over the manuscript in a repetitive way: e.g. the information on the BADH2 gene appears first in line 86 (page 2), then in line 44 (page 7), same applied for other genes. It is recommended to merge these information.
Response: Thanks for your suggestion to improve our manuscript. The suggested changes have been made accordingly and highlighted in the MS in red colour. The knock out of Badh2 gene information has been added (Page 6; Line 211-13) and repeated information has been merged.
3. Line 86: Golden rice was generated through introgression of only two genes: a phytoene synthase and a (bacterial) phytoene desaturase. Introgression of a lycopene cyclase is not required for b-carotene biosynthesis and a b-carotene desaturase does not exist. Paine et al. 2005 (Nat. Biotechn.) as an improvement should be cited. Please correct.
Response: Thanks for your comment. The suggested correction has been made and suggested citation have been made at appropriate place (Citation 110; Page 10; Line 269-71).
4. Most of the genetic engineering approaches are from classical transformation approaches, not from gene editing. Often it is just mentioned that similar results were obtained. So, why CRISPR/Cas9 is the better choice some more arguments and pointing towards convincing comparisons would be good!
Response: Thanks for your valuable suggestion. I agree with you, we have rephrased the title and have had made major changes in the MS to prove advantages of CRISPR/Cas9 system over old techniques. The Figure 2 is elaborated to make the theme of present review clearer.
5. Line 99: avoid subjective expressions (breath-taking)
Response: Thanks for your valuable suggestion. The correction has been made accordingly (“breath-taking to impressive”) and highlighted in red colour.

Reviewer 2 Report
The review does not provide a significant contribution to the field. More specifically, in a special issue about gene editing in plants.
The title does not agree with the content of the article: If the authors say that CRISPR/Cas9 is a better choice for rice grain quality improvement, they have to show that it is really the case. It is not shown in this manuscript, CRISPR/Cas9 method is not discuss, there is only a small paragraph on applications of CRISPR/Cas9 and a non-exhaustive table of edited genes in Oryza sativa, not particularly targeted to genes involved in rice grain quality.
The authors describe a quantity of genes involved in the rice grain quality, but without analyzing which pathway could be the better or the most important to be edited.
A list of genes that have been edited with success in rice are mentioned but protocols and results are not detailed and no critical analysis has been done.
Moreover, other articles from 2018, also describing these genes, were not mentioned in the article:
Bandyopadhyay, A., Yin, X., Biswal, A., Coe, R., & Quick, W. P. (2018) CRISPR-Cas9-Mediated Genome Editing of Rice Towards Better Grain Quality. Rice Grain Quality, 311–336.doi:10.1007/978-1-4939-8914-0_18).
Bao J (2018) Genes and QTL for Rice Grain Quality Improvement. In Rice - Germplasm,Genetics an Improvement. Ed. IntechOpen
http://dx.doi.org/10.5772/56621
Bao J (2019) Biotechnology for rice quality improvement. Rice, 443-471
doi :10.1016/b978-0-1811508-4.00014-9.
A review about genome editing in rice has also been published in 2018, which is more complete and provides further analysis of the choice of the nucleases to obtain better mutation yields. It was not mentioned in the article.
Mishra R, Joshi RK, Zhao K (2018) Genome editing in rice: recent advances, challenges, and future implications. Frontiers in Plant Science 9, 1361.
The authors provide pretty but extremely general figures, which do not provide important information to solve the problem of grain quality.
-Fig 1-Extremely general scheme but not exhaustive, which provide nothing to the improvement of rice quality by gene editing
Typography in the scheme is too small and illegible (for ex in the black oval).
T-DNA instead of t-DNA-The legend is not complete….
-Fig 2-Very general scheme not explained and without interest –The legend is :Key figure ! Key of what?
-Fig 3-Beautiful scheme but without scientific interest, typography too small and incomplete legend. Why to list the genes inside a rice grain? A table could be sufficient because genes and functions are not related to each other or in relation with the different parts of the grain.
Table 1 is a table that provides some references of rice genome editing but the list is non exhaustive and edited genes are not necessarily connected to rice grain quality.
For ex:
Ref 57 is genome editing to induce cold tolerance… ref (56) is not the good ref, it is not the ALS that it is targeted and this ref. have no relation with grain quality.
It would be better to give exhaustive analysis and target more grain quality.
Author Response
Dear editor,
Thank you very much for giving us time and comments regarding our manuscript. Our manuscript “CRISPR/Cas9 is a better choice for rice grain quality improvement: A review” has been revised carefully and here we have given the list of response to the reviewer’s comments. We have improved the manuscript according to the reviewer’s comments. All the revisions have been highlighted with green background color in the revised manuscript.
We are grateful to you and your team for your co-operation in advance.
With best regards,
Sajid Fiaz,
*********************************************************************
Response to reviewers:
Reviewer #2
1. The title does not agree with the content of the article: If the authors say that CRISPR/Cas9 is a better choice for rice grain quality improvement, they have to show that it is really the case. It is not shown in this manuscript, CRISPR/Cas9 method is not discuss, there is only a small paragraph on applications of CRISPR/Cas9 and a non-exhaustive table of edited genes in Oryza sativa, not particularly targeted to genes involved in rice grain quality. The authors describe a quantity of genes involved in the rice grain quality, but without analyzing which pathway could be the better or the most important to be edited. A list of genes that have been edited with success in rice are mentioned but protocols and results are not detailed and no critical analysis has been done.
Response: Thanks for your suggestion to improve our manuscript. The title of manuscript has been revised as per content of the article. The mechanism of CRISPR/Cas9 has been incorporated in the introduction section (Page 2; Line 53-70). The genome edited gene in rice have been grouped into three categories i.e., Quality improvement, Yield Improvement and Disease resistant. The starch biosynthesis pathway has been included along with the information of gene or enzymes involved. And all the revisions have been highlighted with red text in the revised manuscript.
2. Moreover, other articles from 2018, also describing these genes, were not mentioned in the article:
1) Bandyopadhyay, A., Yin, X., Biswal, A., Coe, R., & Quick, W. P. (2018) CRISPR-Cas9-Mediated Genome Editing of Rice Towards Better Grain Quality. Rice Grain Quality, 311–336.doi:10.1007/978-1-4939-8914-0_18).
2) Bao J (2018) Genes and QTL for Rice Grain Quality Improvement. In Rice - Germplasm, Genetics an Improvement. Ed. Intech Open. http://dx.doi.org/10.5772/56621
3) Bao J (2019) Biotechnology for rice quality improvement. Rice, 443-471
doi :10.1016/b978-0-1811508-4.00014-9.
4) Mishra R, Joshi RK, Zhao K (2018) Genome editing in rice: recent advances, challenges, and future implications. Frontiers in Plant Science 9, 1361.
Response: Thanks for your suggestion to improve our manuscript. The suggested citations are included in the introduction section of the revised manuscript. Citation numbers are 17, 18, 19 and 20 (Page 2; Line 71-73). The cited place is highlighted with red colour in text.
3. The authors provide pretty but extremely general figures, which do not provide important information to solve the problem of grain quality. -Fig 1-Extremely general scheme but not exhaustive, which provide nothing to the improvement of rice quality by gene editing. Typography in the scheme is too small and illegible (for ex in the black oval). T-DNA instead of t-DNA-The legend is not complete….
Response: Thank you for your valuable comment for the improvement of figures. The Figure 1 has been revised whereas, it only focused on the basic flow chart of CRISPR/Cas9 system, not grain quality improvement. However, the typography has been fixed with mentioned correction from t-DNA to T-DNA.
3. -Fig 2-Very general scheme not explained and without interest –The legend is: Key figure! Key of what?
Response: Thanks for indicating an important point and we have revised figure 2 to figure 3 and have made changes of scientific interest. The whole CRISPR/Cas9 system has been described along with comparison of old breeding techniques and its utilization in rice grain quality improvement. Additionally, an overview of CRISPR/Cas9 system in functional genomics to improve the grain quality has also been included. The legend “key figure” was meant to describe the application of CRISPR/Cas9 system in rice grain quality improvement which was also the main theme of current review article. However, as per your concern, the word “key figure” has been removed.
4. Fig 3-Beautiful scheme but without scientific interest, typography too small and incomplete legend. Why to list the genes inside a rice grain? A table could be sufficient because genes and functions are not related to each other or in relation with the different parts of the grain.
Response: Thanks for your good comment. We agree with your point of view but, the genes were just put inside the illustration to make them eye catching for the reader. The typography and legends have been revised.
5. Ref 57 is genome editing to induce cold tolerance… ref (56) is not the good ref, it is not the ALS that it is targeted and this ref. have no relation with grain quality.
Response: Thanks for your suggestion. The mentioned correction has been made accordingly (Citation 87; Sun et al., 2016).
6. It would be better to give exhaustive analysis and target more grain quality.
Response: Thanks for your nice suggestion. We have revised every section of the manuscript and grain quality has been described as per your kind suggestion. All the changes are highlighted in red colour in text.

Round 2
Reviewer 2 Report
Report2
Thank you very much to the authors for the revised version which greatly improves the article and thank you for following the recommendations. Figures are more readable and more interesting and the title better matches the content of the article. Legends of the figures are more complete.
However, I still propose some modifications :
In Figure 1 (1b): sgRNA could be omitted from the left box (promoter) because it is duplicative with the right box.
In Figure 1 (5): In the different transformation methods, I don’t understand the difference between protoplast transformation and electroporation & transformation. In my opinion, it is the same thing. So, electroporation & transformation should be omitted.
Furthermore, it was better to write Agrobacterium rhizogenestransformation instead of hairy root transformation and Agrobacterium tumefacienstransformations instead of Agrobacteriumtransformation
In Figure 2 : the genes should be written in italics
In Figure 3 :
tolerant instead of tolrent (two times),
efficiency instead of efficiecy
In the box phenomics : characterization instead of charactrization
In the legend : CRISPRi (CRISPR interference )instead CRISPR interference
L 277,278, 279. Interesting information but without reference ? Add Yang et al 2017
Yang X., Chen L., Yu W. 2017, Knocking out of carotenoid catabolic genes in rice fails to boost carotenoid accumulation, but reveals a mutation in strigolactone biosynthesis. Plant Cell Rep 36, 1533-1545
L282 : The authors must highlight the interest of a multiplex editing if single targets did not yield the expected results.
The authors should also add that CRISPR/Cas9 is the best method to rapidly validate the function of genes and thereby test many genes. It can be used to establish proofs of concept, before targeting the genes that are directly involved in the quality of the rice grain.
Author Response
Dear editor
Thank you very much for giving us time and comments regarding our manuscript. Our manuscript “CRISPR/Cas9 is a better choice for rice grain quality improvement: A review” has been revised carefully and here we given the list of response to the reviewers’ comments. We have improved the manuscript according to the reviewers’ comments. All the revisions have been highlighted with green background color in the revised manuscript.
Again, thanks for your co-operation and expecting further good comments (if any) from you and the two reviewers will be highly appreciated.
With best regards,
Sincerely yours,
Sajid Fiaz,
*********************************************************************
Response to reviewers:
Reviewer #2
1. In Figure 1 (1b): sgRNA could be omitted from the left box (promoter) because it is duplicative with the right box.
In Figure 1 (5): In the different transformation methods, I don’t understand the difference between protoplast transformation and electroporation & transformation. In my opinion, it is the same thing. So, electroporation & transformation should be omitted.
Furthermore, it was better to write Agrobacterium rhizogenestransformation instead of hairy root transformation and Agrobacterium tumefacienstransformations instead of Agrobacteriumtransformation
Response: Thanks for your suggestion to improve our manuscript. All the suggested changes have been made in the figure.
2. In Figure 2: the genes should be written in italics.
Response: Thanks for your valuable suggestion to improve figure. The suggested correction has been made in the figure 2.
3. In Figure 3:
tolerant instead of tolrent (two times),
efficiency instead of efficiecy
In the box phenomics: characterization instead of charactrization
In the legend: CRISPRi (CRISPR interference) instead CRISPR interference Response: Thank you for your valuable comment for the improvement of figure. All highlighted typo corrections are made accordingly.
3. L 277,278, 279. Interesting information but without reference? Add Yang et al 2017
Yang X., Chen L., Yu W. 2017, Knocking out of carotenoid catabolic genes in rice fails to boost carotenoid accumulation, but reveals a mutation in strigolactone biosynthesis. Plant Cell Rep 36, 1533-1545
Response: Thanks for valuable suggestion for the improvement of citation. The suggested citation has been made (Citation # 115).
4. L282: The authors must highlight the interest of a multiplex editing if single targets did not yield the expected results.
The authors should also add that CRISPR/Cas9 is the best method to rapidly validate the function of genes and thereby test many genes. It can be used to establish proofs of concept, before targeting the genes that are directly involved in the quality of the rice grain.
Response: Thanks for your valuable comment and suggestion. We agree with your point of view and to support the multiplex genome editing system some literature has been added (Page # 11; Line # 283-93; Citation # 66, 73, 112 and 113) Moreover, the suggested sentence has been incorporated accordingly (Line # 293-96).
